# Electrochemical Biosensors for Detection of MicroRNA as a Cancer Biomarker: Pros and Cons

**DOI:** 10.3390/bios10110186

**Published:** 2020-11-20

**Authors:** Maliana El Aamri, Ghita Yammouri, Hasna Mohammadi, Aziz Amine, Hafsa Korri-Youssoufi

**Affiliations:** 1Laboratory of Process Engineering & Environment, Faculty of Sciences and Techniques, Hassan II, University of Casablanca, B.P.146, Mohammedia 28806, Morocco; maliana.elaamri@etu.fstm.ac.ma (M.E.A.); ghita.yammouri-etu@etu.univh2c.ma (G.Y.); hasna2001fr@yahoo.fr (H.M.); 2Université Paris-Saclay, CNRS, Institut de Chimie Moléculaire et des Matériaux d’Orsay (ICMMO), Equipe de Chimie Biorganique et Bioinorganique (ECBB), Bât 420, 2 Rue du Doyen Georges Poitou, 91400 Orsay, France; hafsa.korri-youssoufi@universite-paris-saclay.fr

**Keywords:** microRNA, electrochemical biosensor, catalysts, RedOx indicator, cancer biomarker

## Abstract

Cancer is the second most fatal disease in the world and an early diagnosis is important for a successful treatment. Thus, it is necessary to develop fast, sensitive, simple, and inexpensive analytical tools for cancer biomarker detection. MicroRNA (miRNA) is an RNA cancer biomarker where the expression level in body fluid is strongly correlated to cancer. Various biosensors involving the detection of miRNA for cancer diagnosis were developed. The present review offers a comprehensive overview of the recent developments in electrochemical biosensor for miRNA cancer marker detection from 2015 to 2020. The review focuses on the approaches to direct miRNA detection based on the electrochemical signal. It includes a RedOx-labeled probe with different designs, RedOx DNA-intercalating agents, various kinds of RedOx catalysts used to produce a signal response, and finally a free RedOx indicator. Furthermore, the advantages and drawbacks of these approaches are highlighted.

## 1. Introduction

Cancer has been the focus of intense scientific research in recent decades because it includes more than 14 million new cancer cases as well as 8.2 million deaths annually, which makes it one of the most fatal diseases in the world. It is a complex disease characterized by abnormally large cell proliferation, or malignant tumor, formed from the transformation by mutation or genetic instability of an initially normal cell [1]. Attacking some tumor cells that are the source of the disease requires early diagnosis to control and treat them.

The detection of cancer in the early stage of its evolution greatly increases the chances of the treatment success [2]. Indeed, it is based on screening, and on educating patients about early diagnosis. There are many methods of detecting cancer, but the challenge is whether these tests are used to help identify cancer and give an appropriate treatment at an early stage. Among these methods, the most effective ones include imaging exams [3,4] including, radiography, echography, computed tomography scan, magnetic resonance imaging, and positron emission tomography. However, biochemical methods based on the detection and quantification of biomarkers could give an early diagnosis. Biomarkers are defined as substances found naturally in the cells, tissues, or fluids of the human body and present in abnormal amounts in people with cancer or a precancerous condition [5]. Cancer biomarkers could be specific to a single type of cancer or associated with more than one type of cancer [6]. Various cancer biomarkers are known and some of the associated ones with cancer diagnosis are summarized in Table 1.

In recent decades, researchers have focused on microRNA (miRNA) as a cancer biomarker. According to Scopus, and over the years, miRNAs have generated a high amount of interest in the cancer research area, as described in Scheme 1. The number of papers related to miRNA detection as a cancer biomarker was about 650 in 2019, showing the high interest in miRNA analysis.

The interest in targeting miRNAs as cancer biomarkers is related to their biochemical properties and their large amount in biological fluids; they allow easy detection avoiding sample treatment complications.

MiRNAs are small non-coding single-stranded sequences constituted of 18–25 nucleotides [28]. The expression level of the miRNAs is strongly correlated with the onset and development of diseases, including cancer, diabetes, and heart disease [29,30].

The conventional methods used for the quantification and identification of miRNAs are real-time quantitative polymerase chain reaction (RT-qPCR), DNA microarray, Northern blot techniques, and deep sequencing [31]. In general, they have good sensitivity and high specificity, but the methods are complex and need a high level of technology that requires costly equipment and materials, qualified personnel for the assay, and is time consuming [32].

For this reason, the development of other more efficient and less expensive emerging techniques is important and vital for cancer diagnosis and therapy. A variety of methods providing high sensitivity and specificity, and that are easy to handle have been recently developed based on various direct detection methods such as photoelectrochemical, localized surface plasmon resonance, and electrochemical biosensors [33].

Electrochemical biosensors present interest because they can be easily miniaturized, and allow mass-production at a low cost. They could be modified with various recognition elements and are greatly used as versatile devices for nucleic acids-based biosensors development (E-DNA). In addition, such biosensors have demonstrated convincing results with versatile approaches based on newly developed materials and nanomaterials, natural organic and bioorganic polymers, electroactive molecules, catalysts, and biocatalysis, etc. [34,35,36].

In recent years, reviews of electrochemical miRNA biosensors at various viewpoints have been published. Therefore, various strategies based on multi-functional nanomaterials in miRNA biosensors were reported [37,38]. For example, Chen et al. [31] discussed the use of nanomaterials and oligonucleotides as amplification strategies for miRNA detection. Besides, Michael et al. [39] focused more on electrochemical biosensors based on using various combinations of oligonucleotide strategies. Furthermore, Mohammadi et al. [40] reviewed the various amplification strategies based on nanomaterials, oligonucleotides, and enzymes for miRNA analysis and their different possible combinations.

In this review, we provide a compressive overview of various detection approaches of miRNA detection in the literature from 2015 to 2020 following their prevalence. The review focuses on direct miRNA electrochemical biosensing systems based on the RedOx marker. These approaches are based on the RNA biosensors with (i) an electroactive labeled DNA probe sequence, (ii) the use of a catalyst that generates RedOx specie, (iii) the system with DNA RedOx intercalating agent, (iv) the employment of free RedOx indicator and finally, other methods of detection free of the RedOx marker. Various detection approaches are discussed in terms of analytical performances, particularly sensitivity and limit of detection (LOD). We highlight the advantages and drawbacks of these detection approaches. The challenges and successes of these assays are discussed.

## 2. Electrochemical Biosensor Based on Electroactive Labeled Probe Sequence

Electroactive species-labeled DNA probe sequence strategies are widely used for miRNA detection. They provide direct RedOx current response related to the signal variation after the miRNA hybridization. These electroactive species can be inorganic molecules as well as organic ones. For example, metals such as gold nanoparticles [41], silver nanoparticles [42], cadmium [43], and plomb [44] were employed as an inorganic RedOx probe. Organic molecules including, thionine (Thi) [45], ferrocene (Fc) [46], and methylene blue (MB) [47] were mostly used. Indeed, the RedOx molecule could be labeled to the probe sequence directly or with the help of a linker. These methods of labeling will be detailed in this section.

### 2.1. Direct Labeling

Biosensors with direct current response readout are based on short strand DNA probes complementary to the target sequence (miRNA) labeled with molecules of RedOx activity. These labeled probes are firstly immobilized at the surface of the electrode through surface chemistry. The recognition of the miRNA target sequence generates a variation in signal response of the RedOx molecule, which is proportional to miRNA concentration. The literature data show that the labeled DNA probes could be presented with different types of architecture (Figure 1) providing a decrease (ON-OFF signal) or increase (OFF-ON signal) in current response after the hybridization step, starting with a basic design (Figure 1A) and progressing to other more advanced designs based on the elimination of the labeled probe sequence (Figure 1B), the use of a secondary probe labeled with RedOx molecule (Figure 1C) and finally, the use of a two-probe labeled sequence (Figure 1D).

#### 2.1.1. Basic Design

This method is based on the use of a labeled capture DNA probe, which is immobilized at the electrode surface and labeled with a RedOx molecule at the other extremity. However, as shown in Figure 1A, the presence of target miRNA after a hybridization reaction leads to a variation of the distance from the RedOx molecule to the electrode surface, which hampers electron transfer and leads to a decrease in current response (ON-OFF systems). Following this approach, Jou et al. [48], immobilized hairpin DNA probes on a screen-printed carbon electrode (SPCE) modified with AuNPs. The hairpin was labeled at the 5’ end position with MB as a RedOx signal reporter. Before hybridization, the MB is near the electrode surface and thus the electron transfer between the MB and the electrode is possible, so the oxidation response of MB is very high. However, in the presence of the miRNA target, and after displacement amplification reaction and duplex-specific nuclease (DSN), the hairpin was opened indicating the presence of the target, which generates a distance between the RedOx molecule and the surface resulting in a decrease in MB oxidation signal. This method showed an LOD of 3.57 fM for miRNA-155 as a blood cancer biomarker. To improve the sensitivity of detection, Miao et al. [49] used a DNA hairpin immobilized directly on a gold surface electrode and marked with MB at the 5’ end. They introduced the tris (2-carboxyethyl) phosphine hydrochloride reducer (TCEP) providing an enhanced electrochemical signal of MB. The oxidation signal enhancement is based on the activation of MB by reducing its oxidized form in the presence of TCEP. This method allows an improvement of the LOD where 3.2 aM is demonstrated. Yammouri et al. [47] demonstrated a biosensor for miRNA cancer markers with LOD of 1 fM using a basic design where MB-labeled DNA probe sequences and well adapted surface chemistry based on pencil carbon were used. This allows a signal of readout after DNA hybridization without using TCEP as a mediator. The aforementioned methods based on ON-OFF systems lead to an increase in the error of the measurement at low miRNA concentrations. To overcome this problem, a specific architecture of DNA such as loop DNA or DNA stream could be used and the hybridization, in this case, brings the RedOx marker closer to the surface leading to an increase in current response (OFF-ON systems). Wang et al. [50] relied on the “OFF-ON” systems offering a femtomolar detection limit of miRNA-122. In this case, a triple-stem DNA-labeled MB was used. This structure locks MB away from the electrode, and thus blocks electron transfer pathways. Indeed, the signal is turned on only upon the recognition with the target miRNA, which leads to a significant signal response of MB after the hybridization reaction.

#### 2.1.2. Response Based on the Elimination of the Labeled Probe

In this strategy, the labeled probes are released after their hybridization with miRNA targets as shown in Figure 1B. Various methods were employed to achieve this approach. The more popular method is the use of a cleaving agent such as endonucleases [51], duplex-specific nuclease [52], or calcium ions [53] to remove the hybridized duplexes and eliminate the labeled probe from the electrode surface.

The removal could also be obtained by desorption of dsDNA bearing the labeled probe after the hybridization reaction of miRNA due to lower affinity of dsDNA–RNA to the surface (Figure 2). In this case, biosensors based on carbon nanomaterials were demonstrated such as graphene oxide [42] or SWCNTs [46] where the adsorbed ssDNA probe has more affinity to the nanomaterial than dsDNA–RNA does.

Another method uses the competition of miRNA target with two DNA probes in which one is non-labeled and could be hybridized with an MB-labeled duplex reporter. When miRNA target is present, its hybridization reaction with a non-labeled attached probe takes place leading to the displacement of the labeled reporter from the surface [54].

A 2D DNA nanoprobe (DNP) and enzyme-free-target-recycling amplification method based on toehold-mediated strand displacement reactions (TSDRs) was also used to achieve the release of Fc-labeled DNA strands in the case of miRNA-21 detection. The method is based on the displacement of the Fc-labeled DNA strands from the glassy carbone electrode (GCE)/gold nanoparticles (AuNPs) surface after TSDRs, resulting in a decrease in the electrochemical signal. The proposed biosensor has an LOD of 0.31 fM and can be regenerated four times [55].

The previous methods described are simple in the design of the biosensors but lead to a decrease in current signal upon miRNA hybridization with “On-OFF systems”. The detection with increasing response signals based on “OFF-ON Signal” was also demonstrated in the case of the release of the labeled DNA probe. Thus, Fu et al. [56] developed a biosensor based on a homogenous system of detection without immobilization of the ssDNA probe, permitting the reduction in the preparation time of the biosensor. Indeed, the biosensor is based on the negatively charged probes marked with MB at the 3’ end and a negatively charged indium tinoxide (ITO) electrode surface where DNA cannot diffuse easily to the surface due to electrostatic repulsion and the low electrochemical signal of MB is detected. After hybridization, the duplex DNA probe-miRNA was cleaved by DSN hybridization and miRNA continue the second cycle of hybridization cleavage; in the end, this released many short fragments of MB-labeled oligonucleotides with less negative charges, which easily allowed the diffusion on the electrode surface, resulting in a high electrochemical signal.

#### 2.1.3. The Use of Secondary Probe Labeled with RedOx Molecule

This strategy is based on the use of two DNA sequences, the first one is a DNA probe immobilized on the electrode surface, which is targeting a part of miRNA, and the second one is a DNA probe labeled with RedOx molecule targeting the other part of miRNA. After hybridization, the secondary probe labeled with the RedOx molecule leads to a positive signal readout. The principal of this strategy is presented in Figure 1C. This approach was demonstrated by Miao et al. [57] with a biosensor in which an miRNA opened a hairpin capture probe that was immobilized on a gold electrode, while a second probe labeled with silver nanoparticles (AgNPs) was hybridized with the capture probe. A Klenow fragment initiates polymerization of the labeled probe and leads to a release of miRNA. The obtained biosensor provides intense electrochemical signals reaching an LOD of 0.4 fM. Other groups used the same method and amplified the signal readout by the use of rolling circle amplification (RCA) and several probes labeled AgNPs. This leads to an increase in the number of AgNPs on the electrode surface reducing the LOD to 50 aM [58]. Otherwise, with the same principle of detection, other nanoparticles could be used such as CdTe quantum dots (CdTe QDs) instead of AgNPs. With CdTe, a biosensor of miRNA with a very low LOD of 33 aM was demonstrated [59].

An interesting approach was demonstrated for the simultaneous detection of miRNAs using the tetrahedron DNA (TDN) nanostructure (Figure 3A) [60]. This biosensor involved an advanced DNA structure such as TDN which is immobilized on the surface and hybridized with DNA circle as a capture probe presenting many target recognition domains. In the presence of two target miRNAs, and with the assistance of two DNA probes, the mimetic proximity ligation assay (mPLA) can be triggered. The last step is the capturing of two labeled DNA probes labeled with Fc and MB which generate two electrical signal responses. The LOD obtained is in an attomolar range for miRNA-21 and miRNA-155 of 18.9 aM and 39.6 aM, respectively.

Another design of an miRNA biosensor was demonstrated by exploiting the small size of the RedOx molecule, which allows them to be closer to the electrode surface, resulting in an intense RedOx signal. This was obtained by sophisticated design strategies based on coupling the strand displacement reaction and catalytic hairpin assembly recycling (see Figure 3B) [61] or by using an isothermal target recycling amplification strategy [62].

#### 2.1.4. Response Based on Two Labeled Probes

This strategy is based on two probes labeled with different RedOx molecules, in which one is placed near the surface electrode and the second is far from the surface. The signal obtained after RNA hybridization leads to a response where one of the RedOx molecules increases because it becomes closer to the surface and the second one decreases according to the concentration of the hybridized DNA. The final response is a ratio between the two responses (Figure 1D). Various biosensors for miRNA detection were developed following this method in addition to the amplification strategy [63] (see Figure 4A). Most of the developed biosensors used ratiometric electrochemical sensors to obtain a reliable experimental result, which is less sensitive to electrode surface conditions, the probe packing density, the environment, and artificial factors [64,65,66,67].

Amplification strategy using exonuclease-assisted target recycling [63] (Figure 4B) or mismatched catalytic hairpin assembly (CHA) [65] can improve the signal-to-background ratio of the sensor. Using this type of ratiometric design, Zhang et al. [64] developed a more original biosensor based on bipedal DNA walkers for the detection of exosomal miRNA-21. Fc and MB RedOx probes produce a final ratiometric signal, allowing them to obtain an LOD up to 67 aM of miRNA-21. The biosensor developed was applied in a real sample and compared with other methods as RT-qPCR and was regenerated five times without signal decrease.

### 2.2. RedOx Molecules Linker to Nanocarriers

This strategy is based on the use of a RedOx molecule labeled to the probe through a linker such, nanomaterial, polymer, and streptavidin and leads to RedOx signal amplification leading to signal ON. This allows the immobilization of a large number of RedOx molecules after a hybridization reaction leading to the high sensitivity of the biosensor (Figure 5).

Nanomaterials can be used as carriers of the RedOx molecule [44]. AuNPs are usually used in this case as they allow easy attachment through the thiol link. For example, AuNPs have demonstrated an amplification transducer signal conjugated to RedOx molecules such as doxorubicin [69] or cadmium sulfide nanoparticles [66], through sulfur interaction. Another example of a biosensor for miRNA-141 was demonstrated by Yuan et al. [70] where AuNPs were conjugated to Thi- and Fc-labeled hairpin capture probes immobilized on the electrode surface. In the presence of miRNA-141, an increase in the Thi signal is observed, causing a decrease in the Fc signal. The AuNPs-conjugated RedOx molecule could be linked to the capture probe sequence via streptavidin (SA). In this regard, Fc-AuNPs-SA can be used as a labeling nanocarrier for a sandwich biosensor structure. In this system, the capture DNA probes are immobilized on GCE modified with graphene oxide-AuNPs through thiol chemistry. After miRNA hybridization, a second biotinylated capture probe is hybridized. The probe bearing a nanohybrid was added for the quantization of hybridized miRNA-21. This biosensor gives a femtomolar LOD and presents good stability (63 days) [71].

Magnetic nanoparticles such as Fe_3_O_4_ were also employed as a reporter for RNA detection. As an example, a biosensor was developed with the objective of simultaneous detection of miRNA-141 and miRNA-21. In this case, two RedOx molecules, Thi and Fc are attached to the magnetic nanoparticle and the captured probe leading to a large number of RedOx molecules attached. In the presence of miRNAs, the hybridization chain reaction was performed, and then the DNA1/Fe_3_O_4_NPs/Thi and DNA2/Fe_3_O_4_NPs/Fc were captured by the formed dsDNA, which generate a large number of magnetic nanoprobes attached to the surface. A magnified response of currents is obtained with good stability for 4 weeks and was applied for the diagnosis of miRNA in human breast cancer cells [72].

Protein such as streptavidin could be also used as carriers of RedOx molecules. Indeed, AuNPs as RedOx molecules were conjugated to a probe sequence through streptavidin for miRNA-106a detection in the case of gastric cancer diagnosis, causing a lower LOD [41].

Most electrochemical biosensors based on electroactive species-labeled probe sequences for miRNA detection of cancer markers published from 2015 to 2020 are reviewed in Table 2 and their various analytical performances are highlighted.

Direct probe labeling is an approach in which the capture probe is ready to use, therefore it facilitates biosensor’s construction and reduces the required time. The linker molecule labeled probe approach permitts amplification of the signal, but needs an additional step, resulting in an increase in the time and the cost of analysis.

In general, the electrochemical biosensor based on an electroactive species-labeled probe sequence is a commonly used method by researchers, due to the high stability and sensitivity. Indeed, these biosensors enable an LOD in femtomolar to attomolar range and are able to detect miRNA in a complex sample. However, most of the work on biosensors support include an additional amplification procedure or use a sophisticated electrode interface to reach high sensitivity. Additionally, this method is considered as a versatile one, permitting the development of various kinds of biosensor design with new properties. For example, it allows ratiometric dual-current signal responses that provide self-calibration. Consequently, this method can reduce the experimentally and environmental dependent factors/interferences and is effective for miRNA detection in a complex matrix. Moreover, the small size of some RedOx molecules and rational design allow them to be near the electrode surface, which could amplify the signal and detect a small amount of the biomarker. The simultaneous detection was also demonstrated with a simple approach where various biomarkers could be detected at the same time. The direct oxidation/reduction of these molecules without the need of adding other reagents or specific temperature conditions allows them to act as a point of care test. Despite, the above-mentioned advantages, there are a risk of contamination with these molecules and the use of toxic molecules such as cadmium as labeling probe sequence should be avoided for environmental pollution. Concerning the stability of the biosensor with such a design, few works describe these aspects and the stability time is demonstrated between 7 and 30 days depending on the design. Thus, more studies should be highlighted to demonstrate the long-term stability and conditions of storing to allow their actual application.

## 3. Electrochemical Biosensor Based on Catalysts

The catalysts including, enzymes, chemical catalysts, and DNAzyme have been used for miRNA detection (Figure 6). Enzymes are biomacromolecules with highly selective catalytic activities. More than 5000 biological processes have been established to be catalyzed by enzymes. Additionally, they can accelerate chemical reactions with tremendously high efficiency and selectivity, typically with 10^10^–10^15^-fold rate enhancements over uncatalyzed chemical reactions.

Nanomaterials as chemical catalysts can also mimic enzyme catalysis by their ability to act in catalytic processes and are potentially viable alternatives for enzymes. Thus, they attract a great deal of interest and have been actively researched over decades. Indeed, nanomaterials have unique physico-chemical properties, including a size comparable with natural enzymes, a high surface/volume ratio, a large number of catalytically active sites on their surface, as well as the availability of multifunctional reactive groups for subsequent modification and functionalization. The high surface/volume ratio and a large number of active sites should result in a high catalytic efficiency [91].

DNAzymes are ssDNA molecules that also exhibit a catalytic activity and are exploited in biology, medicine, and material sciences. Development in this field is related to the many advantages of DNAzymes over conventional protein enzymes, like simpler preparation and thermal stability [92]. In this section, electrochemical biosensors for miRNA detection based on the enzymes, chemical catalysts, and DNAzyme will be discussed.

### 3.1. Enzyme

The use of an enzyme in amplification strategies is widely used for miRNA detection. It includes various enzymes that lead to obtaining electroactive species which could be electrochemically detected. The most employed enzymes are alkaline phosphatase (ALP) and horseradish peroxidase (HRP). The binding strategy of the enzymes to the biosensors is the key to biosensors’ success because it could affect the accessibility of the active site or lead to denaturation. In this regard, different works have been reported based on the different types of binding approach on the biosensor that will be discussed in this part. Some examples of enzyme binding approaches are presented in Figure 6A.

#### 3.1.1. Enzyme–Steptavidin Binding

The binding of an enzyme on the biosensor via the interaction between biotin-modified probe sequence and streptavidin-modified enzyme is a habitually used method in the literature of enzyme binding employed for miRNA detection [93,94,95,96]. Using this type of interaction between a biotinylated capture probe and streptavidin-conjugated alkaline phosphatase (SA-ALP), Xia et al. [97] described an electrochemical–chemical–chemical (ECC) RedOx cycling system for miRNAs detection with Fc methanol, which acts as the RedOx mediator using TCEP as a reducing agent and L-ascorbic acid 2-phosphate as a subtract of ALP enzyme. With this system, one ALP enzyme captured by one target miRNA molecule favored the production of thousands of ascorbic acid (AA) enzymatic products, which allowed a sensitive detection with an LOD of 40 fM. This biosensor could be generated 10 times, which permits an increase in the sample throughput and reduces the sample analysis time. In another work based also on the ECC system, a sandwich biosensor was employed using a biotinylated DNA-linked GO-AuNPs hybrid as a signal probe, which was interacted with SA-ALP. A capture probe was immobilized on GCE/AuNPs/Magnesium oxide (MgO) for more sensitive miRNA detection of 50 aM [98]. Mandli et al. [99] developed an electrochemical miRNA biosensor based also on a sandwich system with the use of AuNPs as a biosensor platform that incorporates SA-ALP enzyme linked to a biotin-modified signaling probe, catalyzing α-naphthyl phosphate as a substrate to produce electroactive α-naphthol. The differential pulse voltammetry (DPV) technique was used for the measurement of α-naphthol oxidation.

A competitive RNA/RNA hybridization assay-based biosensor was developed by incorporation of SA-HRP linked to biotinylated capture probes for amperometric detection of miRNA using H_2_O_2_ as enzyme-substrate and hydroquinone (HQ) as RedOx mediator. Indeed, different platforms—one based on the screen-printed electrode (SPE)/AuNPS [100] and the other one based on GCE/tungsten diselenide/AuNPs [101]—were employed. A lower LOD of 0.06 fM was obtained using tungsten diselenide/AuNPs because tungsten diselenide displayed a large effective surface area. This allowed increased loading of AuNPs on its surface to act as an excellent sensing substrate, which therefore can immobilize more capture probe DNA to the electrode and in turn leads to a low LOD.

Using a new way for probe capture immobilization, Torrente-Rodríguez et al. [102] employed magnetic-beads (MBs) modified with a special DNA–RNA antibody as capture probe bioreceptor. The antibody recognized the hybridized microRNA and biotinylated capture probe linked to SA-HRP. Indeed, amperometric detection implying the H_2_O_2_/HQ system at disposable SPCE was performed. This methodology has been evaluated for the quantification of miRNA-205 and miRNA-21 in total RNA obtained from human breast tissues.

In another work, simultaneous detection of four miRNAs using DNA tetrahedral nanostructure-based sandwich-type assay and Poly-HRP40 was performed in a serum sample of pancreatic cancer patients. A biotinylated capture signaling probe linked to HRP-SA was hybridized with the immobilized DNA tetrahedral on the gold electrode surface. Indeed, the HRP enzyme catalyzed the reduction in H_2_O_2_ in the presence of microRNAs, with TMB employed as an electron mediator, and thus generated a quantitative amperometric signal in the presence of TMB substrate [103]. Despite, the advantages of simultaneous detection with the presented method, it is still unable to give a specific LOD of each analyzed miRNA because of the interaction of the enzyme with all of the biotinylated signal probe complementary sequences of each miRNA.

Enzyme reaction-based electrochemical biosensor-integrated hybridization chain reaction (HCR) amplification was used to enhance the sensitivity of detection. Using this method, a large amount of streptavidine enzyme binds to biotin labeled on the long-range HCR product, which could remarkably amplify electrochemical signals. Different electrochemical biosensor based HCR techniques and the interaction between biotin-modified probe sequence and streptavidin-modified enzyme were developed [104,105]. Indeed, Zhai et al. [104] developed an electrochemical biosensor based on HCR and ALP by measuring the oxidation of α-naphthol as the enzymatic product, which is proportional to the miRNA concentration, an LOD of 0.56 fM was obtained. This biosensor has a good stability of 2 weeks.

Otherwise, in other work, DSN was employed; indeed, biotinylated ssDNA capture probes were immobilized on gold electrodes allowing SA-ALP to be attached to the capture probe, facilitating the production of an electrochemically active p-aminophenol (p-AP) from p-aminophenyl phosphate (p-APP) substrate. The resulting p-AP was cycled by TCEP after its electro-oxidization, enabling an increase in the anodic current of p-AP, which is proportional to miRNA concentration. Due to the cleavage of the double-strand DNA (dsDNA) by DSN, a decrease in p-AP response was observed after hybridization indicating the presence of miRNA [106]. More sensitive biosensors were developed also using DSN and the introduction of nanomaterials as a biosensor platform. However, the detection of miRNA was performed by the evaluation of AA using ALP and ECC RedOx cycling, using different platforms such as GCE/molybdenum disulfide (MoS_2_)/AuNPs and GCE/MWCNTs@graphene oxide nanoribbons (GONRs)/AuNPs [107,108].

Other works for miRNA detection based on enzymatic reaction reported the use of CHA, which is initiated by the presence of target miRNA [109,110,111,112]. CHA allows the infinite recycling of miRNA to create a mass of streptavidin-enzyme modified signal probes, leading to an enhancement of electrochemical response. Additionally, to further enhance the sensitivity for miRNA detection, nanomaterials linked to enzymes were used. In this regard, Chen et al. [109] proposed a sandwich system integrating CHA and carbon sphere-MoS_2_ (CS-MoS_2_)/AuNPs for the immobilization of capture DNA (see Figure 7). AuNP-modified biotinylated signaling probes were employed as carriers of Avidin-HRP, which catalyze the H_2_O_2_ + HQ system to produce a strong electrochemical response, used for sensitive miRNA detection, to achieve a lower LOD of 0.16 fM.

Zhang et al. [113] used DSN for miRNA-21 detection coupled with a capture probe labeled with biotin and SA-coated AuNPs, which were immobilized on the electrode surface due to the SA/biotin interaction. The numerous SA-coated AuNPs can subsequently immobilize a large number of biotin-labeled HRP molecules. Indeed, AuNPs were used as nanocarriers for HRP, helping to maintain their enzymatic activity. HRP was used to catalyze the reduction in H_2_O_2_ and in the presence of TMB, electrochemical current signals can be generated, which considerably enhances the electrochemical signal for miRNA-21 detection. The proposed biosensor permits an LOD of miRNA-21 down to 43.3 aM. The proposed biosensor allows an analysis of miRNA-21 in the human lung cancer cell line (A549 cells).

#### 3.1.2. Enzyme–Protein Binding

The immobilization of enzymes is also possible by a protein. In this regard, Fang et al. [114] reported an electrochemical biosensor based on the immobilization of enzyme HRPs through zinc finger protein, which binds preferably to the DNA−RNA hybrid formed between an ssDNA capture probe and a target miRNA-21. For sensitive detection, ECC was used as an amplifier system based on the induction of a series of oxide-reduction reactions in the presence of HRP including Ru(NH_3_)_6_^3+^/Ru(NH_3_)_6_^2+^, BQ/HQ, and TCEP. The LOD for miRNA-21 in the buffer and diluted human serum were 2 and 30 fM, respectively.

Enzyme-conjugated protein was also employed for labeling an antibody that specifically recognizes DNA–RNA duplex [115]. In this view, Vargas et al. [116] involved the use of direct hybridization of miRNA-21 with a specific biotinylated DNA probe immobilized on streptavidin-modified MBs. A specific antibody labeled with a bacterial protein A conjugated with Poly-HRP40 recognized the capture probe/miRNA-21 duplex. Amperometric detection of miRNA-21 was performed upon the magnetic capture of the modified MBs onto the SPCEs using the H_2_O_2_/HQ system.

Zouari et al. [117] developed an electrochemical AuNP-based biosensor platform that was used for miRNA detection. Indeed, RNA/miRNA homoduplexes were recognized with the viral protein p19, labeled with a HRP-conjugated anti-maltose binding protein monoclonal antibody (see Figure 8). The bioplatforms present at least 2 months of storage stability. Additionally, the analysis of miRNA in total RNA extracted from healthy and cancerous breast cells was performed using the proposed biosensor.

#### 3.1.3. Other Types of Enzyme Binding

There is another way for enzyme immobilization at the electrode surface for miRNA detection. Indeed, graphene quantum dots (GQDs) have a large surface-to-volume ratio, excellent compatibility of GQDs were used as a new platform for a large amount of HRP immobilization through the non-covalent assembly (see Figure 9). In this work, a sandwich system was employed for miRNA-155 detection integrating capture probe and signaling probe modified with NH_2_ conjugated with QDs-HRP. The proposed biosensor is permitted to reach an LOD of 0.14 fM in a linear range from 1 fM to 100 pM [118]. Otherwise, an electrochemical biosensor for miRNA-221 detection using CHA and also a sandwich system based on the use of HRP directly labeled to signal probe was reported. HRP was used to catalyze the reaction of TMB/H_2_O_2_ for amperometric detection of miRNA-221 [119].

### 3.2. Chemical Catalysts

Different kinds of nanomaterials as chemical catalysts, including copper-based metal–organic framework (Cu-MOF), copper nanoclusters (CuNCs), Fe_3_O_4_NPs, platinum/tunable tin-doped indium oxide nanoparticles (Pt/Sn-In_2_O_3_) were utilized for miRNA detection and will be discussed in this part. An example of an electrochemical biosensor for miRNA detection based on the chemical catalyst is presented in Figure 6B.

Wang et al. [120] proposed a paper modified AuNP as a biosensor platform for miRNA-155 detection using a capture probe-AuNPs@Cu-MOF (see Figure 10). In the presence of glucose, Cu-MOFs and AuNPs as chemical catalysts cooperatively catalyzed the glucose oxidation, resulting in the wide linear detection range from 1.0 fM to 10 nM and the LOD of 0.35 fM for miRNA-155. The present biosensor showed stability of 30 days.

In other work, CuNCs as catalyst-based biosensor were employed for miRNA detection. CuNCs were synthesized at the electrode surface by taking DNA–RNA heteroduplexes as templates with the help of AA and Cu^2+^. Besides, the formed CuNCs possessed the capability of catalyzing H_2_O_2_ reduction, resulting in steady and amplified electrochemical signals, which were used in miRNA analysis, reaching an LOD of 8.2 fM [121]. As other types of nanomaterial as a chemical catalyst, SA/Pt/Sn-In_2_O_3_ hybrids were employed for miRNA-21 biosensor development. Indeed, a SA/Pt/Sn-In_2_O_3_ was attached to a biotinylated hairpin capture probe by the presence of miRNA-21, producing an electrochemical signal of oxygen reduction for detection in O_2_-saturated solution. SA/Pt/Sn-In_2_O_3_ as the amplifier led to an LOD of 1.92 fM [122].

Otherwise, a mixing of enzyme and nanomaterial as enzyme–nanomaterials composite was reported to enhance the sensitivity for miRNA detection [123,124,125,126]. This is obtained by the synergetic effect of the catalyst and nanomaterials that improve the electrochemical response. However, the LOD of miRNAs obtained with this composite could be also obtained using just one type of catalyst; consequently, the use of enzyme and nanomaterials as catalysts together increases the cost and the time of biosensor construction without significant signal amplification.

### 3.3. DNAzyme

Recently, DNAzymes with peroxidase-like activity has aroused a high interest. To achieve such catalytic activity, the DNA probe requires a G-quadruplex structure, which is able to bind to hemin molecules. Such a system promotes a RedOx reaction between the target molecule and hydrogen peroxide, leading to the formation of an oxidized product. Indeed, hemin-G-quadruplex was employed for miRNA detection [127,128]. The principle of an electrochemical biosensor based on hemin-G-quadruplex as a DNAzyme is presented in Figure 6C. In this context, hemin-G-quadruplex was employed to catalyze H_2_O_2_ reduction, with coupling with HCR to fabricate long hemin-G-quadruplex DNAzyme nanowires (see Figure 11) [129]. In other work, hemin-G-quadruplex was also used for miRNA analysis by catalyzing the oxidation of TMB in the presence of H_2_O_2_ [130].

Electrochemical biosensors based on catalysts with the focus on their various analytical performances are summarized in Table 3.

Compared to DNAzyme and nanomaterials, enzymes generally display superior performance in terms of catalytic efficiency and specificity, as well as excellent activity in aqueous media under ambient conditions, thanks to their high turnover and high selectivity. On the other hand, the enzyme is influenced by the external environment, requires a specific reaction condition, the reaction time dependent on enzyme activity and finally we always need an additional molecule (substrate) for the detection of the enzymatic product. Otherwise, nanomaterials and DNAzyme present desired advantages including their high stability, a tunable structure, catalytic efficiencies, purely synthetic routes to their preparation, lower cost, and excellent tolerance to experimental conditions. Despite the above-mentioned advantages, there is a risk of contamination with these molecules.

## 4. Electrochemical Biosensor Based on RedOx Intercalating Agent

In chemistry, intercalation is the insertion of a molecule (or a group of molecules) between two other molecules (or groups). In the present approach, a RedOx molecule or a complex of RedOx molecules are employed for DNA strand binding via intercalation. The binding to ssDNA and dsDNA is obtained with different affinity regarding the nature of the intercalator. Different types of molecules were used as RedOx intercalating agents for miRNA analysis. This includes organic molecules such as the commonly used MB [136] as well as oracet blue (OB) [137] and toluidine blue [138]. Various other molecules and macromolecules could also be intercalated as an organometallic complex including Ru(NH_3_)_6_^3+^ [139], cobalt phenanthroline, or metal intercalating agents such as palladium nanoparticles [140] and biomolecules as hemin [141]. The principle of the approaches is presented in Figure 12. The interlacing process on the biosensor could be direct to the target DNA strands (Figure 12A) [138] or indirect by the design of specific sites in the DNA probe called a template [142] (Figure 12B).

### 4.1. Direct Intercalation

The direct intercalating of RedOx molecules between the DNA strands is based on an interaction of electroactive species on the formed double-strand DNA–DNA or DNA–RNA upon hybridization reaction (Figure 12A). Indeed, various kinds of RedOx molecules could be used in the case of electrochemical biosensors for miRNA that will be discussed in this part and they are divided into three kinds, electroactive molecules, and electroactive complex.

#### 4.1.1. Electroactive Molecule

The MB is the popular electroactive indicator used as an intercalating DNA agent since MB interacts both with ss and ds DNA where the binding mode differs from each macromolecule. Thus MB and dsDNA interact by three binding modes: (i) intercalation between successive base pairs with the face-to-face binding of the bases and MB; (ii) insertion into the minor groove; and (iii) insertion into the major groove of the double helix. The interaction ssDNA with MB is obtained by face-to-face binding of the bases and MB. Thus, the affinity of MB regarding ssDNA and dsDNA is largely discussed. Regarding the literature, depending on the biosensors design, MB shows a higher affinity for ssDNA or dsDNA. In the case of miRNA detection, most biosensors published show less affinity of MB to DNA–RNA complex. Thus, it has been reported by Li et al. [143] that MB reacts easily with guanine present in ssDNA with high affinity due to the good accessibility of MB to the guanine of ssDNA. In this work, a transducer formed with GCE/MWCNTs/PAMAM modified with a DNA probe was used as a biosensors platform for miRNA-24 detection. The detection was followed through measuring the electroactivity of MB before and after hybridization. A decrease in the signal response is observed after the hybridization of miRNA-24 where LOD reached 0.5 fM. Furthermore, this biosensor was stable for 7 days and could be regenerated four times.

Nevertheless, a more complex biosensor design was performed using RedOx intercalator and amplification strategy in order to enhance the sensitivity of the detection. For example, a biosensor was developed for miRNA-155 detection, where HCR was used to amplify the number of intercalated MB molecules on the DNA strands, by creating a longer dsDNA and using a GCE modified with polypyrrole/reduced graphene oxide/AuNPs (Ppy/rGO/AuNPs) [144]. The amplification of MB response can also be performed by using a 3D DNA nanonet structure which is hybridized with the immobilized capture probe on a gold electrode by a sandwich system. In this case, a femtomolar detection of miRNA-21, was obtained [145]. Both presented biosensors were stable for 2 weeks and the analysis of miRNA was performed in spiked human serum.

Another approach involving nanomaterials as nanocatalysts which assisted the signal amplification strategy has been used to enhance the RedOx signal of MB intercalator. For example, the association of various kinds of nanomaterials as a catalyst was developed to improve the MB response. Thus, the synergetic effect of Fe_3_O_4_ and cerium dioxide (CeO_2_) decorated with gold nanoparticles (Fe_3_O_4/_CeO_2_@AuNPs) was demonstrated to improve the RedOx response of MB intercalator (Figure 13A). A labeled probe decorated with the nanocatalysts was employed in a sandwich system for the amplification of intercalated MB response by a direct catalyzation of MB reduction leading to LOD of 0.33fM [146]. Other nanomaterials such as carboxylate-reduced graphene oxide (COOH-rGO) have been reported as an amplification strategy. The nanomaterial can intercalate on the ssDNA, leading to an accumulation of electroactive MB (Figure 13B). In the presence of miRNA, a DSN cleave captures probe/miRNA duplex is obtained resulting in a decrease in MB response. A lower LOD of 0.01 fM was obtained [147].

Other approaches involving various innovative amplification strategies have also been published. For example, Guo et al. [148] reported an attomolar biosensor for the detection of miRNA-196a. The mechanism consists of after hybridization of miRNA and formation of a terminal deoxynucleotidyl, transferase will trigger a DNA extension reaction producing long ssDNA rich in guanine, in which MB is attached. The biosensor presents a very low sensitivity compared to other ones, used with other strategies due to obtained long ssDNA which can specifically adsorb positively charged MB via guanine bases, resulting in the attachment of a large amount of MB. However, this biosensor needs many steps after miRNA hybridization which can limit their application.

#### 4.1.2. Electroactive Metals Complex

In the case of electroactive metal complex intercalator, hexaammineruthenium III chloride (RuHex) is used frequently as electroactive complex employed for miRNA detection. The RuHex is positively charged and could intercalate with DNA strand via the binding with the anionic phosphate of DNA through electrostatic interaction. Few studies have been published with this complex and the associated amplification strategies. For example, hierarchical flower-like gold nanostructures (HFGN) were developed as a biosensor platform and used for selective detection of miRNA-21 in the buffer and real sample; RuHex was employed as RedOx intercalator [149]. In other works and to increase the amount of intercalated RuHex, Yu et al. [150] and Chen et al. [151] used DNA nanostructures as probes immobilized on the gold electrode surface for enhancing the binding ability of RuHex. Thus, in the presence of miRNAs targets, these nanostructures provided an enhancement of RuHex intercalants. The analysis of miRNA in total RNA-extracted cancerous breast cells was performed using the proposed biosensors.

### 4.2. Intercalation via Template

The intercalation via template consists of the use of other molecules, which need to create recognition sites in DNA strands for their intercalation. The principal of this type of intercalation is presented in Figure 12B. Generally, the nanostructure of DNA is necessary to perform this method.

Starting with hemin that needs abundant guanines to form hemin/G-quadruplexes [152], an electrochemical biosensor was described by Wang et al. [153] for the quantification of miRNA-21 at fM level. The catching of hemin was enhanced by the integration of a sandwich system in which a signal probe is labeled with N-doped graphene/Au nanoparticles (NG-AuNPs) and works as a support for several strands of guanine-rich DNA.

Otherwise, Zhang et al. [140] described a label-free and attomolar electrochemical miRNA-21 biosensor based on a template for palladium nanoparticles witch lead to integration with the nitrogen of guanine. RCA amplification was used to produce a massive G-rich long ssDNA resulting in an enhanced electrochemical signal.

### 4.3. Other Type of Intercalation

Some other RedOx molecules able to intercalate poorly or that do not intercalate at all with the DNA strands have also been studied. However, their intercalation could be realized via a linker. In this regard, Asadzadeh et al. [154] used AgNPs as a RedOx molecule, which was intercalated to the ssDNA via single-walled carbon nanotube (SWCNT). The binding of the AgNPs/SWCNT nanohybrid to ssDNA was performed via interactions π–π between the nanohybrid and the nitrogenous bases of ssDNA. The proposed method was used for miRNA-25 detection as a lung cancer biomarker.

Wang et al. [155] used GO as support of Prussian blue (PB), which is the RedOx intercalant agent. The GO was adsorbed on the 3′ end of capture probe though π–π interaction. Then, the PB was attached to GO. In the presence of miRNA-122, the GO with the assembled PB was separated from the electrode surface due to the low affinity of the GO with the DNA/RNA hybrid, resulting in a decrease in electrochemical response of PB.

MiRNAs are characterized by cis-diol at the 3′-terminal, this propriety is employed by Liu et al. [156] for an attomolar detection of miRNA-21 using AgNP as a RedOx intercalate agent. Indeed, with the presence of miRNA-21, a 4-mercaptophenylboronic acid (MPBA) was attached in the 3′-terminal of miRNAs through the boronate ester bond formation and then captured AgNP via the Ag–Thiol interaction. Meanwhile, free MPBA molecules in solution induced the in situ assemblies of AgNPs on the electrode surface via the covalent interactions between α-hydroxycarboxylate of citrate and boronate of MPBA and the formation of Ag–Thiol bonds.

Electrochemical biosensors based on RedOx intercalating agents highlighting their various analytical performances are presented in Table 4.

A comparison between different methods of intercalation used for miRNA detection shows that intercalation via a template presents some advantages such as good biocompatibility, good electrochemical properties facilitating a very low LOD of miRNA detection. Nevertheless, it still presents some limitations related to the complicated process of fabrication, which is considered time-consuming; and also, the need for an additional amplification step to obtain high sensitivity which is required for miRNA detection. Otherwise, direct intercalation of the RedOx molecules method is considered as an easy and fast method of intercalation since the used RedOx molecules could bind directly and specifically to DNA without the need of complicated preparation. Indeed, the LOD obtained using this method is slightly higher than the ones obtained with intercalation via the template method.

Overall, the intercalation strategy is easy to use on-site, but it does not allow simultaneous detection, because the interaction of the RedOx molecule is not specific and can intercalate on all DNA strands present at the surface of the electrode. Furthermore, the use of intercalation based on electrostatic interaction leads to a non-specific interaction and high background noise, especially using real samples.

## 5. Electrochemical Label-Free Biosensing

Label-free biosensor include the use of ferri–ferrocyanide complex or hexaammineruthenium (II)/(III) as RedOx-free indicators. The response is based on electrostatic repulsion or interaction depending on the marker. The ferri–ferrocyanide complex is negatively charged, thus after the hybridization of miRNA target, a repulsion effect is produced by the negatively charged phosphate leading to the variation of response. In the case of hexaammineruthenium (II)/(III) complex, it is positively charged and could undergo interaction with hybridized DNA. Another factor that could lead to the variation of the RedOx marker is its accessibility to the surface after DNA–RNA complex formation of a duplex preventing electron transfer to the surface. Various electrochemical methods could be used to follow such responses including electrochemical impedance spectroscopy (EIS), cyclic voltammetry (CV), DPV, and square wave voltammetry (SWV). The principle of electrochemical biosensors based on the free RedOx indicator is presented in Figure 14. In this section, electrochemical biosensors for miRNA detection based on the free RedOx indicator will be discussed.

### 5.1. Ferri/Ferrocyanide as Free RedOx Indicator

Ferri–ferrocyanide as a free RedOx indicator was widely employed for miRNA detection. The detection is based on the electrostatic repellence of two negatively charged DNA and Fe(CN)_6_^3−/4−^ molecules leading to the decrease in electrochemical reaction on the surface upon hybridization and decrease in current response or increase in impedance (Figure 14). In this regard, EIS as the detection method has been used extensively for miRNA monitoring in combination with various materials and nanomaterials as transducers [163,164,165,166]. In most research, the nature of materials attached to the surface plays an important role in the electrochemical response. In this respect, Yammouri et al. [167] used this approach in the association of transducer formed with a pencil graphite electrode (PGE) modified with a carbon black-bearing DNA probe. The miRNA-125a detection was monitored by EIS in the presence of this RedOx marker. The synergetic effect of negatively charged carbon black combined with the high surface ratio of PGE allows the detection with lower LOD and good determination in serum samples. In another work, the polythiophene film-modified screen-printed gold electrode was employed as a biosensor platform for miRNA-221detection from total RNA extracted from human lung and breast cancer cell lines. The proposed biosensor demonstrated the benefic effect of the conductive surface [168]. Mandli et al. [169] employed PGE modified with Ppy for microRNA-34a detection by EIS. Indeed, the immobilization of the probe was performed during Ppy electropolymerization on PGE and the hybridization was performed by the specific recognition sequence of miRNA-34a. This biosensor was functional for the analysis of miRNA-34a in human breast cancer cells samples.

Association of this method with system amplification such as HCR was also performed. Indeed, Meng et al. [170] developed an electrochemical biosensor combining efficient HCR for signal amplification of oligonucleotides with negatively charged repelling [Fe(CN) _6_]^3−/4−^ ions inducing a spatial blockage to the electron transfer. In this biosensor, many linear DNA concatamers lead to a great increase in interfacial charge-transfer resistance (R_ct_), which is positively correlated with miRNA-21 concentrations with an LOD of 4.63 fM with the stability of the biosensor lasting for 21 days. This strategy allowed the analysis of miRNA-21 in different cancer cells including breast, cervical, and non-small cell lung cancers.

Zhang et al. [171] employed a magnetic bead-modified glassy carbon electrode combined with a DSN amplification strategy for impedimetric miRNA-21 detection (see Figure 15). Due to the cleavage of the capture probe–miRNA-21 heteroduplex, after the hybridization steps the negatively charged layer could not be formed, resulting in a small R_ct_ in the presence of ferriferrocyanide, which was used for miRNA-21 detection, permitting an LOD of 60 aM.

Otherwise, DPV based on ferriferrocyanide was also employed for miRNA detection [172,173,174]. In this case, advanced surface modification was performed to obtain an efficient electron transfer. Indeed, a biosensor formed with fluorine-doped tin oxide electrode modified with nanomaterials composed with nitrogen-doped functionalized graphene associated with (AgNPs and polyaniline (PANI) nanocomposite modified, was developed for miRNA detection. The employed nanocomposite allowed more biomolecules to be immobilized at the surface of the electrode, which shortened the distance for electron transfer and ion diffusion paths from the capture probe to the nanomaterials. The nano-biosensor showed a wide dynamic detection range of 10 fM–10 μM and a low LOD of 0.2 fM [175].

Ferriferrocyanide as a free RedOx indicator was used for other strategies of detection based on the direct adsorption of miRNAs on the electrode surface. This strategy needs a preliminary step of target isolation mostly with capture probe-modified magnetic beads, then, a denaturation of the DNA–RNA hybrid is achieved by heating at 95 °C. Thereafter, researchers try different ways for adsorbing isolated miRNAs. For instance, a picomolar biosensor was reported by Boriachek’s group based on gold electrode–miRNA affinity interaction [176]. Wan’s group used a screen-printed graphene electrode, for miRNAs detection isolated and directly adsorbed on the surface of the graphene electrode via graphene–miRNA affinity interaction. This method showed an LOD of 10 fM [177]. According to the present two works, we can conclude that graphene–miRNA affinity is higher than the gold–miRNA affinity. Koo et al. [178] magnified the adsorption of miRNAs using a polyadenine extension, which has a high affinity with the gold surface. However, the miRNA was subjected to poly (A) extension on 3′ ends using poly(A) polymerase enzyme.

### 5.2. Hexaammineruthenium (II)/(III) Chloride as Free RedOx Indicator

Hexaammineruthenium (II)/(III) was also employed for miRNA detection. In this regard, nickel phosphate nanostructures (NiPNs) were used as a biosensor platform for the immobilization of the capture probe by coordination bonding between Ni and probe DNA especially phosphate groups. The constructed NiPNs-p-DNA surface acted as the amplified platform enabling efficient access to many target miRNA-21 sequences. The probe-DNA immobilization and the miRNA hybridization steps were supervised by EIS measurements in [Ru(NH_3_)_6_]^3+/2+^. The proposed biosensor allowed reaching an LOD of 0.034 pM, and the analysis of miRNA-21 levels in human lung cancer cells [179].

Association of the two RedOx indicators ferricyanide and hexaammineruthenium (II)/(III) for miRNA detection was also explored. In this view, a ratiometric electrochemical miRNA biosensing platform based on the target-triggered ruthenium release and RedOx recycling was reported (see Figure 16). In this research, [Ru(NH_3_)_6_]^3+^ was entrapped into the pores of mesoporous silica nanoparticles modified by an indium tin oxide electrode and was subsequently capped by a capture probe. Once the target miRNA was captured and hybridized into dsDNA/RNA, [Ru(NH_3_)_6_]^3+^ was released and electroreduced into [Ru(NH_3_)_6_]^2+^, which was then chemically oxidized back to [Ru(NH_3_)_6_]^2+^ by [Fe(CN)_6_]^3−^. The consumed [Fe(CN)_6_]^3−^ and liberated [Ru(NH_3_)_6_]^3+^ produced a significant ratiometric signal. Using this innovative approach, an LOD that decreased down to 33 aM was obtained. Additionally, the developed biosensor showed good stability over 20 days and permitted the analysis of miRNA-21 in different cancer cells including breast, cervical, and non-small cell lung cancers [180].

Electrochemical biosensors based on free RedOx indicator with the focus on their various analytical performances are exposed in Table 5.

The discussed methods in this section present some advantages including, higher sensitivity, easier signal quantification, direct conversion of the biological event into an electrical signal, and finally requiring fewer steps of fabrication since there is no need to use RedOx markers that interact with DNA strands. On the other hand, the inconvenience is that these methods suffer from some drawbacks such as the potential of non-specific adsorption of other biomolecules on the electrode surface, which may cause false-positive interference. Additionally, the results of this method were always easily disturbed by surface contamination and adsorption.

## 6. Other Methods of MicroRNA Electrochemical Detection

Other methods were employed in the electrochemical biosensor area for the detection of miRNAs as biomarkers of cancer, including the guanine oxidation method, RedOx current of the electrode surface, and labeled miRNA.

### 6.1. Oxidation of Guanine

The guanine is a purine nucleobase present in miRNAs strands, which have RedOx active groups employed in the construction of the electrochemical biosensors. The approach of following the oxidation of guanine presents the first approach used in the case of a DNA biosensor and could be used also for miRNA detection. Akbarnia et al. [187] proposed enzymatic digestion biosensors for femtomolar detection of miRNA-541as lung cancer biomarkers. Indeed, the probe is immobilized on the pencil graphite electrode modified graphene quantum dots (GQDs/PGE). Meanwhile, in the presence of miRNA-541, Hinf1 as a restriction enzyme cleaved the formed capture probe–miRNA-541 duplex. The oxidation of guanine was measured in the presence and absence of miRNA-541. After hybridization, a decrease in guanine’s electrochemical signal was observed, because the DNA strands containing guanine were removed by the enzyme. In another use of the guanine oxidation method, Azab et al. [188] described a very sensitive biosensor at a zeptomolar level for miRNA let7-a detection using complementary sequence capture probe free-guanine bases. The capture probe was immobilized on the carbon paste electrode/carbon nanotubes/chrysine/gold nanoparticles (CPE/CNT/C/AuNPs) platform. The CNT/C film increases the surface area of the biosensor platform, increasing the conductivity, and thus is responsible for signal amplification.

The current method is very simple but has some limitations, especially considering that the oxidation of guanine bases as a free molecule is easier than in DNA strands and so could generate an unreliable result. As a matter of fact, the integration of a probe sequence without guanine, and its replacement with inosine that does not have the same potential for oxidation of guanine, is necessary in order to obtain only guanine oxidation response after the hybridization step (OFF-ON signal). Therefore, because the binding between cytosine and guanine is stronger than the binding of guanine with inosine, the use of such sequence does not provide a higher level of binding between the probe and miRNA compared to the sequence of the guanine-continuing DNA probe. Moreover, this method does not allow the regeneration of the biosensor because the oxidation of guanine is irreversible.

### 6.2. RedOx Current from Electrode Surface

Electrochemical biosensors designed for the detection of miRNAs based on RedOx current from the electrode surface were dismantled as a detection method. This method consists of modifying the electrode surface with an electroactive molecule before immobilizing the capture. The detection of miRNAs is done by electrochemical monitoring of the response of the redox molecule deposited on the electrode surface before and after the hybridization probe [189,190,191]. For this purpose, a simple model has recently been described by Zouari et al. [192] for the quantification of miRNA-21 at fM level, using screen-printed carbon electrode/pyrene carboxylic acid/rGO/AuNPs as biosensor platform. A 6-ferrocenylhexanethiol (Fc-SH) as a RedOx molecule was immobilized on the electrode surface.

### 6.3. Labeled MicroRNA

This method needs a labeled miRNA, which provides a hard step for the biosensor application in real samples. A labeled miRNA biosensor was recently described by Sabahi et al. [193] for the quantification of miRNA-21 at fM level. This was obtained by the use of cadmium ions (Cd^2+^) which is linked to a phosphate group of miRNA via an electrostatic reaction. Then, the labeled miRNA hybridize with a capture probe immobilized on a fluorine-doped tin oxide electrode/SWCNTs/dendritic gold nanostructures through to Au–thiol interaction.

## 7. Conclusions and Future Perspectives

This work reviewed the progress in the development of electrochemical miRNAs biosensors using different approaches based on an electroactive species-labeled probe sequence, catalyst, RedOx intercalating agent, RedOx system, among others. In view of the various published papers, all miRNA strategies of detection based on electrochemical biosensors discussed in this review are presented according to their percentage of use (Figure 17). This distribution indicated clearly that the RedOx indicator as labeled, intercalant, or free RedOx indicator is widely used for miRNA detection with a percentage of 69% compared to catalytic detection, which presents 26%. In general, all approaches discussed in this review have almost equal use for miRNA detection, except the categories of other methods, which are still not developed yet compared to their employment in DNA biosensors. This is due probably to the lack of the association of these methods with the amplification approach generally used in the case of miRNA detection.

The synthesis study clearly showed that electrochemical biosensors are efficient and practical approaches towards the analysis of miRNAs in the clinical field with high sensitivity. Nevertheless, the problem of miRNA analysis in the real sample cannot be neglected regarding the intercalating agent RedOx and the free RedOx indicator. These approaches have certain limitations, including the likely occurrence in the real sample of interferences, which could affect the results obtained. Otherwise, approaches based on electroactive species labeled with a probe sequence and a catalyst enable an accurate and precise analysis of miRNAs in a real sample. In addition, simultaneously detecting two miRNAs is more favorable with the RedOx-labeled probe sequence strategy, compared to the other strategies discussed. Although the LODs of the discussed biosensors are very low, most of the biosensors have not been tested and validated with a large number of samples from cancer patients and control groups. On the other hand, given the fact that, in general, known miRNAs are not specific to a single pathology, particular attention must also be paid to the development of electrochemical biosensors dedicated to the simultaneous quantification of a group of miRNAs to facilitate cancer diagnosis with improved reliability.

Although electrochemistry stands out due to its inherent miniaturization, mass production, and low cost, there are still significant challenges to meet before portable, robust, user-friendly point-of-care biosensors for cancer diagnosis through the detection of circulating miRNA expression profiles become a reality. Furthermore, because of these challenges, to date, to the best of our knowledge, no commercial electrochemical biosensor for circulating miRNA analysis is available; consequently, further effort should be devoted to validation, clinical assays, and the commercialization in the near future.

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
