# Peer review of "Electrochemical Biosensors for Detection of MicroRNA as a Cancer Biomarker: Pros and Cons"

_biosensors, 2020, doi:10.3390/bios10110186_

Round 1

Reviewer 1 Report

November 2nd, 2020.

Ms. Leira Hao

Assistant Editor

Biosensors Editorial Office

Correspondence reference: biosensors-996264-peer-review-v1

“Electrochemical Biosensors for Detection of MicroRNA as a Cancer Biomarker: Pros and Cons”

Dear Editor,

The manuscript, “Electrochemical Biosensors for Detection of MicroRNA as a Cancer Biomarker: Pros and Cons” reviewed the progress in the development of electrochemical miRNAs biosensors using different approaches based on electroactive species labeled probe sequence, catalyst, RedOx intercalating agent, RedOx system, and others. The authors have explained several strategies to detect microRNA and illustrated with some examples. The review is well done and well written. I recommend minor revision this manuscript version with special attention for:

  1. Lines 68-69: “MiRNAs are small non-coding single-stranded sequences of rebounding nucleotides acids in the order of 18-25 [27].” Please, check the unit of measurement: 18-25 what?
  2. Lines 137-138: “Following this approach, Jou et al., immobilized hairpin DNA probe on screen-printed carbon electrode (SPCE) modified with AuNPs.”. Please add a reference for Jou et al.
  3. Please, check if the references have not been forgotten.
  4. Lines 145-147: “To improve the sensitivity of detection, Miao et al. used a DNA hairpin immobilized directly on a gold surface electrode and marked with MB at the 5' end.” Please add a reference for Miao et al.
  5. Line 150: “Yammouri et al. demonstrated”. Reference for Yammouri.
  6. Line 153: “TCPE” or TCEP?
  7. Line 158: reference for Wang et al.
  8. Line 189: reference for Fu et al.
  9. Line 205: reference for Miao et al.
  10. Line 253: reference for Zhang et al.
  11. Line 270: reference for Yuan et al.
  12. Line 367: reference for Xia et al.
  13. Line 377: reference for Mandli et al.
  14. Line 390: reference for Torrente-Rodríguez et al.
  15. Line 410: reference for Zhai et al.
  16. Line 435: reference for Chen et al.
  17. Line 441: reference for Zhang et al.
  18. Line 451: reference for Fang et al.
  19. Line 459: reference for Vergas et al.
  20. Line 469: reference for Zouari et al.
  21. Line 496: reference for Wang et al.
  22. Line 595: reference for Li et al.
  23. Line 630: reference for Guo et al.
  24. Line 656: reference for Wang et al.
  25. Line 660: reference for Zhang C. et al.
  26. Line 667: reference for Asadzadeh et al.
  27. Line 671: reference for Wang et al.
  28. Line 676: reference for Liu et al.
  29. Line 733: reference for Yammouri et al.
  30. Line 740: reference for Mandli et al.
  31. Line 746: reference for Meng et al.
  32. Line 753: reference for Zhang et al.
  33. Line 779: reference for Koo et al.
  34. Line 833: reference for Akbarnia et al.
  35. Line 860: reference for Zouari et al.
  36. Line 866: reference for Sabahi et al.
  37. The manuscript is well written.
  38. The literature citation needs more updates. Following reference must be included in the manuscript to make it better: Genosensor made with a self-assembled monolayer matrix to detect MGMT gene methylation in head and neck cancer cell lines (https://doi.org/10.1016/j.talanta.2019.120609).
  39. The Figures are well-done and well-showed. Beautiful work!

Author Response

The manuscript, “Electrochemical Biosensors for Detection of MicroRNA as a Cancer Biomarker: Pros and Cons” reviewed the progress in the development of electrochemical miRNAs biosensors using different approaches based on electroactive species labeled probe sequence, catalyst, RedOx intercalating agent, RedOx system, and others. The authors have explained several strategies to detect microRNA and illustrated with some examples. The review is well done and well written. I recommend minor revision this manuscript version with special attention for:

We thank the reviewer for the compliments. The manuscript has been revised accordingly. The answer to comment raised by the reviewer is bellow.

  1. Line 68: “MiRNAs are small non-coding single-stranded sequences of rebounding nucleotides acids in the order of 18-25 [27].” Please, check the unit of measurement: 18-25 what?

Response: We thank the reviewer for his/her comments. The unit of measurement was checked and this sentence was rephrased accordingly,

Line 68: «MiRNAs are small non-coding single-stranded sequences of rebounding nucleotides acids in the order of 18-25 [27] » is replaced by « MiRNAs are small non-coding single-stranded sequences constituted of 18-25 nucleotides [28]. ».

All the mentioned references are now added after the author’s name in the revised manuscript. Please note that line numbers are changed in the revised manuscript.

  1. Lines 137-138: “Following this approach, Jou et al., immobilized hairpin DNA probe on screen-printed carbon electrode (SPCE) modified with AuNPs.”. Please add a reference for Jou et al

Line 135:  Jou et al. [48] …

  1. Please, check if the references have not been forgotten.

All these references are listed in the reference section.

  1. Lines 145-147: “To improve the sensitivity of detection, Miao et al. used a DNA hairpin immobilized directly on a gold surface electrode and marked with MB at the 5' end.” Please add a reference for Miao et al.

Line 144: Miao et al. [49]…

  1. Line 150: “Yammouri et al. demonstrated”. Reference for Yammouri.

Lines 148 - 149: Yammouri et al. [47]…

  1. Line 153: “TCPE” or TCEP?

Response: we thank the reviewer for his/her remark. The correct one is TCEP. It was corrected in the revised version (line 151).

  1. Line 158: reference for Wang et al.

Line 156 : Wang et al. [50]…

  1. Line 189: reference for Fu et al , …

Line 187 : Fu et al. [56]…

  1. Line 205: reference for Miao et al.

Line 203 - 204 : Miao et al. [57]…

  1. Line 253: reference for Zhang et al.

Line 251 : Zhang et al. [64]…

  1. Line 270: reference for Yuan et al.

Line 268 : Yuan et al. [70]…

  1. Line 367: reference for Xia et al.

Line 366 - 367 : Xia et al. [97]…

  1. Line 377: reference for Mandli et al.

Line 376 : Mandli et al. [99]…

  1. Line 390: reference for Torrente-Rodríguez et al.

Line 389 : Torrente-Rodríguez et al. [102]…

  1. Line 410: reference for Zhai et al.

Line 409 : Zhai et al. [104]…

  1. Line 435: reference for Chen et al.

Line 432 : Chen et al. [109]…

  1. Line 441: reference for Zhang et al.

Line 438 : Zhang et al. [113]…

  1. Line 451: reference for Fang et al.

Line 448 : Fang et al. [114]…

  1. Line 459: reference for Vergas et al.

Line 456 : Vergas et al. [116]…

  1. Line 469: reference for Zouari et al.

Line 466 : Zouari et al. [117]…

  1. Line 496: reference for Wang et al.

Line 493 : Wang et al. [120]…

  1. Line 595: reference for Li et al.

Line 588 : Li et al. [143]…

  1. Line 630: reference for Guo et al.

Line 622 : Guo et al. [148]…

  1. Line 656: reference for Wang et al.

Line 648 : Wang et al. [153]…

  1. Line 660: reference for Zhang C. et al.

Line 652 : Zhang et al. [140]…

  1. Line 667: reference for Asadzadeh et al.

Line 659: Asadzadeh et al. [154]…

  1. Line 671: reference for Wang et al.

Line 663: Wang et al. [155]…

  1. Line 676: reference for Liu et al.

Lines 668 - 669 : Liu et al. [156]…

  1. Line 733: reference for Yammouri et al.

Line 725 : Yammouri et al. [167]…

  1. Line 740: reference for Mandli et al.

Lines 731 - 733 : Mandli et al. [169]…

  1. Line 746: reference for Meng et al.

Line 738 : Meng et al. [170]…

  1. Line 753: reference for Zhang et al.

Line 745 : Zhang et al. [171]…

  1. Line 779: reference for Koo et al.

Line 771 : Koo et al. [178]…

  1. Line 833: reference for Akbarnia et al.

Line 823 : Akbarnia et al. [187]…

  1. Line 860: reference for Zouari et al.

Line 859 : Zouari et al. [192]…

  1. Line 866: reference for Sabahi et al.

Line 85661 : Sabahi et al. [193]…

Taking into account the above comment of the reviewer, we correct also the following references:

Line 90 : Chen et al. [31]…

Line 91 : Michael et al. [39]…

Line 93 : Mohammadi et al. [40]…

Line 830 : Azab et al. [188]…

  1. The manuscript is well written.

Response: We thank the reviewer for the compliments.

  1. The literature citation needs more updates. Following reference must be included in the manuscript to make it better: Genosensor made with a self-assembled monolayer matrix to detect MGMT gene methylation in head and neck cancer cell lines (https://doi.org/10.1016/j.talanta.2019.120609).

Reponses: As requested by the reviewer, the following reference https://doi.org/10.1016/j.talanta.2019.120609 was added in Table 1 in the revised manuscript and in the reference section (reference 27).

Replying to your precious comments and to the reviewer 4 comments. Other updates have been done:

Table 3: Reference [133].

Table 3: Reference [134].

  1. The Figures are well-done and well-showed. Beautiful work!

Reponse: Many Thanks.

Reviewer 2 Report

This paper goes over direct miRNA electrochemical biosensors based on the use of redox markers which have been described in literature from 2015 to 2020. The review is extensive, well presented, organized and discussed so I think it only needs minor revision, before publication in Biosensors:

  1. Line 268: reference 190 is mentioned there and also in Table 2. References should be mentioned correlatively, so this reference should have another numeration, 68, and the following ones change numbering consistently.
  2. Again same thing: in line 731 authors jump from reference 161 to reference 175.
  3. Figure 15 legend: authors have jumped from reference 174 to reference 182 and also in this legend some of this reference 182 has been left there.
  4. Shouldn’t the paragraph from line 871 to line 879 be in the Conclusions Section better than where it is now placed?
  5. Check English language: there are some typos and mistakes along the text.

Author Response

This paper goes over direct miRNA electrochemical biosensors based on the use of redox markers which have been described in literature from 2015 to 2020. The review is extensive, well presented, organized and discussed so I think it only needs minor revision, before publication in Biosensors:

Response: We thank the reviewer for the compliments. The manuscript has been revised accordingly. The answer to comment raised by the reviewer is bellow.

  1. Line 268: reference 190 is mentioned there and also in Table 2. References should be mentioned correlatively, so this reference should have another numeration, 68, and the following ones change numbering consistently.

Response: we thank the reviewer for this comment. Please note that in the revised version of our manuscript the references were re-organized accordingly. The reference 190 is now 69 (line 267 and in Table 2) which corresponds to reference 68 in the former version.

  1. Again same thing: in line 731 authors jump from reference 161 to reference 175.

Response: we thank the reviewer for this comment. Please note that in the revised version of our manuscript, all references were organized accordingly. The reference 175 is now 166 (line 723).

  1. Figure 15 legend: authors have jumped from reference 174 to reference 182 and also in this legend some of this reference 182 has been left there.

Response: we thank the reviewer for this comment. Please note that in the revised version of our manuscript all references were organized accordingly. 174 is now 179 and 182 is now 180. (line 781 and line 783).

  1. Shouldn’t the paragraph from line 871 to line 879 be in the Conclusions Section better than where it is now placed?

Response: we thank the reviewer for his/her comment. You are right, the paragraph from line 871 to line 879 is now placed in conclusion section (lines 864 - 872).

Please note that replying to the reviewer 1 and 4 comments, some updates have been added:

Table 1: Ref [27]

Table 3: Ref [133]

Table 3: Ref [134]

Accordingly, these new references have been listed in the reference section.

  1. Check English language: there are some typos and mistakes along the text.

Response: We thank the reviewer for this comment. The manuscript has been revised in the order to improve the English. Indeed, some words were changed and some sentences were rephrased. Please note that also in responding to reviewer 3's comments we have changed some sentences that were misspelled. All these modifications are highlighted in red (in marked revision of the manuscript):

Line 16: “2nd …“

Lines 36 - 37: “The detection of cancer in the early stage of its evolution increases greatly the chances of the treatment success [2]”

Lines 39 – 47: “Among these methods and the most effective ones, imaging exams [3] [4] including, radiography, echography, Computed Tomography scan, magnetic resonance imaging, and Positron Emission Tomography. However, biochemical methods based on the detection and quantification of biomarkers could give an early diagnosis. Biomarkers are defined as substances found naturally in the cells, tissues, or fluids of the human body and present in abnormal amounts in people with cancer or a precancerous condition [5]. Cancer biomarkers could be specific to a single type of cancer or associated with more than one type of cancer [6]. Various cancer biomarkers are known and some of the associated ones with cancer diagnosis are summarized in table 1.

Lines 60 – 62: “The interest of targeting miRNAs as cancer biomarkers is related to their biochemical properties and their large amount in biological fluids. They allow easy detection avoiding sample treatment complications. »

Lines 81 – 86: “Electrochemical biosensors present interest because they can be easily miniaturized, and allow mass-production at a low cost. They could be modified with various recognition elements and are greatly used as versatile devices for nucleic acids based biosensors development (E-DNA). In addition, such biosensors have demonstrated convincing results with versatile approaches based on newly developed materials and nanomaterials, natural organic and bioorganic polymers, electroactive molecules, catalysts, and biocatalysis, etc [34] [35] [36].”.

Lines 88 – 89: “reviews of electrochemical miRNA biosensors at various viewpoints have been published”.

Line 93: “Furthermore, recently Mohammadi et al. [40] reviewed the various… ”.

Line 102: “Various detection approaches are discussed in terms…”.

Line 106: “Electroactive species labeled to DNA probe sequences approach are widely used for miRNA…”.

Line 106: “ it provides…”

Lines 121 – 122: “starting with a basic design (Figure 1.A) to other more advanced where based on the elimination…”.

Line 135: “…to a decrease of current response (ON-OFF systems).”

Lines 146 – 148: “The oxidation signal enhancement is based on the activation of MB by reducing its oxidized form in presence of TCEP.”

Lines 172 – 174: “When miRNA target is present, its hybridization reaction with a non-labeled attached probe takes place leading to the displacement of the labeled reporter from the surface [54]. »

Lines 246 – 247: “two RedOx markers (left) standard…”

Lines 312 – 313: “Indeed, these biosensors permit to achieve a LOD in a range of femtomolar to attomolar range …”

Line 316: “this method is considered as a versatile …”.

Line 317: “it allows  ratiometric… ”.

Line 318 : “Consequently, permitting to reduce the experimental..”.

Line 592: “A decrease of the signal response is observed…”.

Reviewer 3 Report

The review presents an overview of the most recent electrochemical biosensors for miRNA detection focusing on cancer detection. Many approaches regarding miRNA determination based on different strategies such as RedOx DNA-intercalating agents, RedOx-labelled probes, and labeled free RedOx probes are described. The manuscript is well-structured and organized, containing enough information and the most recent and relevant references regarding the selected topic. Even though I recommend its publication in this Journal, I highly encourage the authors to carefully review the English language and correct any mistakes that can be found in the text such as missing “s” in the form of the verb, missing “s” in some words used as plural, the use of wrong words, connectors,… Here there are some examples.

There are several paragraphs where many short sentences are just numbered one after the other, sometimes even they are repetitive. Please review this matter and make use of some connectors instead, since that would make it easier to read the manuscript. Here some examples: abstract, lines 81-87, lines 105-108,

Please, review the use of the expression “The most” along the whole text.

Line 87 “…In recent years, reviews of electrochemical miRNA biosensors at various viewpoints are published…” should be have been published.

Line 101 “…The various detection approach is discussed…. It should be “….Various detection approaches are….”

Line 106 “It´s provided”

Line 121 “…to other more developed ones…” I guess the correct expression here is “…to other more advanced where …..”

Line 137 “…which hamper electron transfer and lead to a decrease of current response (ON-OFF system)….” Missing “s” in the form of the verbs. Please, check for more along the text.

Line 148 “…which is based on the activate MB by reducing its oxidized form and enhance the oxidation RedOx signal…” not clear sentence, Might it be something as follows? “…which is based on the activation of MB by reducing its oxidized form and, thus enhancing the oxidation signal…”

 Line 174 “… When miRNA target is present hybridization reaction with a non-labeled attached probe is obtained leading to the displacement of the labeled reporter from the surface…” It should be “… When miRNA target is present, its hybridization reaction with a non-labeled attached probe takes place leading to the displacement of the labeled reporter from the surface…”

Line 181 “…were also to achieve the release….” There is a missing word here “used”

Line 267 “…AuNPs have demonstrated a signal transducer and could be conjugated to RedOx molecules such…” there is a missing word.

There are a few mistakes in lines 313 to 320.

The quality of Figure 7 can be improved.

Line 459, Vergas et al. should be changed by Vargas et al.

Line 470 “…RNA/miRNA homoduplexes were known with the viral protein…” it should be were recognized or were bound to the protein.

As a suggestion, section 5. Electrochemical Biosensor Based on Free RedOx Indicator, I would rewrite the name of this section since, in my opinion, it is a bit confusing since RedOx probes are used, I would write label-free biosesing.

Author Response

The review presents an overview of the most recent electrochemical biosensors for miRNA detection focusing on cancer detection. Many approaches regarding miRNA determination based on different strategies such as RedOx DNA-intercalating agents, RedOx-labelled probes, and labeled free RedOx probes are described. The manuscript is well-structured and organized, containing enough information and the most recent and relevant references regarding the selected topic. Even though I recommend its publication in this Journal, I highly encourage the authors to carefully review the English language and correct any mistakes that can be found in the text such as missing “s” in the form of the verb, missing “s” in some words used as plural, the use of wrong words, connectors,… Here there are some examples.

Response: Thank you for these remarks. In the revised manuscript the English language was revised and the mistakes was corrected. All these modifications are highlighted in red, for example:

Line 102: “Various detection approaches are discussed in terms…”.

Line 106: “Electroactive species labeled to DNA probe sequences approach are widely used for miRNA…”.

Lines 246 – 247: “two RedOx markers (left) standard…”

Line 316: “this method is considered as a versatile …”

Line 317: “it allows  ratiometric… ”.

There are several paragraphs where many short sentences are just numbered one after the other, sometimes even they are repetitive. Please review this matter and make use of some connectors instead, since that would make it easier to read the manuscript. Here some examples: abstract, lines 81-87, lines 105-108,

Response: Yes, thank you for these remarks, you are right. In the revised manuscript, we have rephrased and connected the sentences, for example:

Lines 81 – 86: Electrochemical biosensors present interest because they can be easily miniaturized, and allow mass-production at a low cost. They could be modified with various recognition elements and are greatly used as versatile devices for nucleic acids based biosensors development (E-DNA). In addition, such biosensors have demonstrated convincing results with versatile approaches based on newly developed materials and nanomaterials, natural organic and bioorganic polymers, electroactive molecules, catalysts, and biocatalysis, etc [34] [35] [36].

Lines 106-109: Electroactive species labeled DNA probe sequences strategy are widely used for miRNA detection. It provides direct RedOx current response related to the signal variation after the miRNA hybridization. These electroactive species can be inorganic molecules as well as organic ones. For example, metals such as gold nanoparticles [41],

Please, review the use of the expression “The most” along the whole text.

Response : We thank the reviewer for his/her remark, you are right the expression “the most’ was repeated in the former version. The sentences were re-phrased in the revised version as follow:

Line 31 “The most fatal…” is replaced by “the fatal…”

Lines 264 – 265 “The most used is AuNPs as they allow easy attachment through thiol link.” is replaced by  “AuNPs are usually used in this case as they allow easy attachment through thiol link.”

Lines 363 – 364 “The binding of an enzyme on the biosensor via the interaction between biotin modified probe sequence and streptavidin modified enzyme is the most used method in the literature of enzyme…” is replaced by “ The binding of an enzyme on the biosensor via the interaction between biotin modified probe sequence and streptavidin modified enzyme is the common method in the literature of enzyme biding employed for miRNA detection [93] [94] [95] [96].

Line 580 “The MB is the most popular electroactive indicator…” is replaced by “The MB is the popular electroactive indicator…”

Lines 587 - 588 “the most biosensors published show less affinity of MB…” is replaced by  “most biosensors published show less affinity of MB…”

Lines 631 - 632 “hexaammineruthenium III chloride (RuHex) is the most used electroactive complex employed for miRNA detection…” is replaced by “hexaammineruthenium III chloride (RuHex) is used frequently as electroactive complex employed for miRNA detection.”

Lines 864 – 865 “the most different miRNA strategies detection based on electrochemical…” is replaced by  “all miRNA strategies detection based on electrochemical…”

Line 87 “…In recent years, reviews of electrochemical miRNA biosensors at various viewpoints are published…” should be have been published.

Line 101 “…The various detection approach is discussed…. It should be “….Various detection approaches are….”

Line 106 “It´s provided”

Line 121 “…to other more developed ones…” I guess the correct expression here is “…to other more advanced where …..”

Line 137 “…which hamper electron transfer and lead to a decrease of current response (ON-OFF system)….” Missing “s” in the form of the verbs. Please, check for more along the text.

Response: We thank the reviewer for these corrections; they are taken in consideration in the revised manuscript (lines- 88 – 89; line 102; line 107; line 122; line 135)

Line 148 “…which is based on the activate MB by reducing its oxidized form and enhance the oxidation RedOx signal…” not clear sentence, Might it be something as follows? “…which is based on the activation of MB by reducing its oxidized form and, thus enhancing the oxidation signal…”

Lines 147 - 148: … They introduce the tris (2-carboxyethyl) phosphine hydrochloride reducer (TCEP) providing an enhanced electrochemical signal of MB.  The oxidation signal enhancement based on the activation of MB by reducing its oxidized form in presence of TCEP.

 Line 174 “… When miRNA target is present hybridization reaction with a non-labeled attached probe is obtained leading to the displacement of the labeled reporter from the surface…” It should be “… When miRNA target is present, its hybridization reaction with a non-labeled attached probe takes place leading to the displacement of the labeled reporter from the surface…”

 Lines 172 - 174:  Another method, use the competition of miRNA target with two DNA probe, one is none labeled and could be hybridized with MB labeled duplex reporter. When miRNA target is present, its hybridization reaction with a non-labeled attached probe takes place leading to the displacement of the labeled reporter from the surface [54].

Line 181 “…were also to achieve the release….” There is a missing word here “used”

Response: of course, thank you for this remark.

Line 179: 2D DNA nanoprobe (DNP) and enzyme-free-target-recycling amplification method based on toehold-mediated strand displacement reactions (TSDRs) were also used to achieve the release of Fc-labeled DNA strands in the case of miRNA-21 detection.

Line 267 “…AuNPs have demonstrated a signal transducer and could be conjugated to RedOx molecules such…” there is a missing word. 

Response: Sentence has been rephrased and the missing word was added.

Lines 265 - 266: …..AuNPs have demonstrated amplification transducer signal conjugated to RedOx molecules such…

There are a few mistakes in lines 313 to 320.

Response : We thank the reviewer for these remarks, this paragraph  is now as following: ;

“In general, the electrochemical biosensor based on electroactive species labeled probe sequence are mainly used method by researchers, due to their high stability and sensitivity. Indeed, these biosensors permit to achieve a LOD in a range of femtomolar to attomolar range and are able to detect miRNA in a complex sample. However, most of the work developed biosensors support an additional amplification procedure or use a sophisticated electrode interface to reach high sensitivity. Additionally, this method is considered as a versatile one, permitting to develop various kinds of biosensor design with new properties. For example, it allows ratiometric dual-current-signal responses that provides self-calibration. Consequently, permitting to reduce the experimental and environmental dependent factors/interferences and is effective for miRNA detection in a complex matrix.” (lines 311 - 320 )

The quality of Figure 7 can be improved.

Response : The figure 7 has been improved. Indeed, the resolution of the figure was enhanced.

Line 459, Vergas et al. should be changed by Vargas et al.

Response : We thank the reviewer for this remark, “Vergas et al.” is now replaced by ‘Vargas et al.” (line 456).

Line 470 “…RNA/miRNA homoduplexes were known with the viral protein…” it should be were recognized or were bound to the protein.

Response : The  sentence “Indeed, RNA/miRNA homoduplexes were known with the viral protein….“ is replaced by “Indeed, RNA/miRNA homoduplexes were recognized with the viral protein… ” (line 467) as suggested by reviewer.

As a suggestion, section 5. Electrochemical Biosensor Based on Free RedOx Indicator, I would rewrite the name of this section since, in my opinion, it is a bit confusing since RedOx probes are used, I would write label-free biosesing.

Response: The section 5. Electrochemical Biosensor Based on Free RedOx Indicator has been rephrased according to the suggestion of the reviewer.

  1. Electrochemical label-free biosensing

Reviewer 4 Report

The manuscript is interesting review dealing with electrochemical biosensors for detection of miRNA. Manuscript is well written, but need clarifying some issues mentioned below before its publication.

Some papers (e.g. Electroanalysis 2019, 31, 293–302) from M. Bartosik group should be cited.  

In table 1, first record contains two times BRCA1 and BRCA2 biomarkers.

What authors mean by “rebounding nucleotides acids” at page 3 line 68?

At page 7 line 238 is word near repeated two times, same at page 9 line 276 with word capture.

At page 8 line 246 is mentioned “figure 4 top”, but this figure does not contain such panel.

Sentence starting at page 32 line (This strategy allowed the analysis of miRNA-21……) is probable unfinished.  

Guanine is not a nucleotide.

Inosine is redox active nucleoside not a base.

Author Response

The manuscript is interesting review dealing with electrochemical biosensors for detection of miRNA. Manuscript is well written, but need clarifying some issues mentioned below before its publication.

Response: We thank the reviewer for the compliments. The manuscript has been revised accordingly. The answer to comment raised by the reviewer is bellow

Some papers (e.g. Electroanalysis 2019, 31, 293–302) from M. Bartosik group should be cited.

Response: As requested by the reviewer, the following references Electroanalysis 2019, 31, 293–302 and Analytical and bioanalytical chemistry, 412(21), 5031-5041 were added in the revised manuscript (Table 3). Ref [133] and Ref [134].

Please note that replying to the reviewer 1, other updates have been added:

Table 1 : Ref [27]

Accordingly, these new references have been listed in the reference section.

In table 1, first record contains two times BRCA1 and BRCA2 biomarkers.

Response: We thank the reviewer for his/ her attention. The repeated BRCA1 and BRCA2 were removed from Table 1.

What authors mean by “rebounding nucleotides acids” at page 3 line 68?

Response: rebounding nucleotides acids means that a stand is rich in nucleotides. The corresponding sentence was rephrased according to the comment of reviewer 1 (line 68).

Line 68: «MiRNAs are small non-coding single-stranded sequences of rebounding nucleotides acids in the order of 18-25 [27]» is replaced by « MiRNAs are small non-coding single-stranded sequences constituted of 18-25 nucleotides [28]. »

At page 7 line 238 is word near repeated two times, same at page 9 line 276 with word capture.

Response: We thank the reviewer for his/ her comment. The repeated words was removed (line 236 and line 274)

At page 8 line 246 is mentioned “figure 4 top”, but this figure does not contain such panel.

Response: We thank the reviewer for his/ her comment. Please note that we changed “top” by “right” and now it is well mentioned (line 241)

Sentence starting at page 32 line (This strategy allowed the analysis of miRNA-21……) is probable unfinished.  

Response: We thank the reviewer for his/ her comment. Please note that in this sentence “ and finally to have a” was added by mistake, now it was removed (lines 742-744)

« This strategy allowed the analysis of miRNA-21 in different cancer cells including breast, cervical, and non-small cell lung cancers. »

Guanine is not a nucleotide.

Inosine is redox active nucleoside not a base.

Response: We thank the reviewer for his/ her comments. According these clarifications, the corresponding sentences were rephrased (lines 820 and lines 837,839).

Line 820 “The guanine is a nucleotide present in miRNAs strands” is replaced by “The guanine is a purine nucleobase present in miRNAs strands”

Lines 837 - 839 “As a matter of fact, the integration of a probe sequence without guanine bases and with a base having no RedOx activity such as inosine is necessary in order to obtain a correct guanine oxidation response after the hybridization step” is replaced by “As a matter of fact, the integration of a probe sequence without guanine, and replaced it with inosine that has not the same potential oxidation of guanine, is necessary in order to obtain guanine oxidation response after the hybridization step (off-on signal)”
